# Relationship between People’s Interest in Medication Adherence, Health Literacy, and Self-Care: An Infodemiological Analysis in the Pre- and Post-COVID-19 Era

**DOI:** 10.3390/jpm13071090

**Published:** 2023-07-01

**Authors:** Andrea Grandieri, Caterina Trevisan, Susanna Gentili, Davide Liborio Vetrano, Giuseppe Liotta, Stefano Volpato

**Affiliations:** 1Department of Biomedicine and Prevention, University of Rome “Tor Vergata”, 00133 Rome, Italy; giuseppeliotta@hotmail.com; 2Geriatric and Orthogeriatric Unit, St. Anna University Hospital of Ferrara, 44124 Ferrara, Italy; caterina.trevisan@unife.it (C.T.); stefano.volpato@unife.it (S.V.); 3Aging Research Center, Department of Neurobiology, Care Sciences and Society, Karolinska Institutet and Stockholm University, 141 86 Stockholm, Sweden; susanna_gentili@virgilio.it (S.G.); davide.vetrano@ki.se (D.L.V.); 4Stockholm Gerontology Center, 141 86 Stockholm, Sweden

**Keywords:** medication adherence, health literacy, self-care, infodemiology, Google Trends

## Abstract

The prevalence of non-communicable diseases has risen sharply in recent years, particularly among older individuals who require complex drug regimens. Patients are increasingly required to manage their health through medication adherence and self-care, but about 50% of patients struggle to adhere to prescribed treatments. This study explored the relationship between interest in medication adherence, health literacy, and self-care and how it changed during the COVID-19 pandemic. We used Google Trends to measure relative search volumes (RSVs) for these three topics from 2012 to 2022. We found that interest in self-care increased the most over time, followed by health literacy and medication adherence. Direct correlations emerged between RSVs for medication adherence and health literacy (r = 0.674, *p* < 0.0001), medication adherence and self-care (r = 0.466, *p* < 0.0001), and health literacy and self-care (r = 0.545, *p* < 0.0001). After the COVID-19 pandemic outbreak, interest in self-care significantly increased, and Latin countries showed a greater interest in self-care than other geographical areas. This study suggests that people are increasingly interested in managing their health, especially in the context of the recent pandemic, and that infodemiology may provide interesting information about the attitudes of the population toward chronic disease management.

## 1. Introduction

In the last few decades, the prevalence of non-communicable diseases has grown exponentially, especially among older people. In parallel, the clinical management of these conditions—and patients’ survival—have made great progress thanks to the development of effective drugs to control symptoms and slow their progression [1]. As a result, the prevalence of non-communicable diseases such as cardiovascular disease, diabetes, cognitive disorders, and chronic respiratory diseases will increase further in the near future, and more individuals will live with chronic conditions in old age. This will result in older people taking more and more drugs chronically, resulting in complex drug regimens [2]. However, about 50% of patients with chronic diseases show poor adherence to the prescribed medications [3,4]. Medication adherence (MA), defined as “the extent to which a patient’s behavior matches the prescribed drug dosing regimen, including time, dosage, and interval of medication intake” [5], is a crucial issue in this context. Indeed, several studies have estimated that poor medication adherence leads to an increase in morbidity and mortality and is estimated to entail costs of about $100 billion per year [6].

Another relevant aspect of chronic disease management that is an essential part of treatment is self-care (SC). Self-care for chronic diseases has been defined as “a process of maintaining health through health promotion interventions and disease management” [7]. In the last decades, several studies have shown that self-care may positively influence several health-related outcomes, such as medication adherence, the need for hospitalizations, quality of life, and survival [8,9]. Self-care maintenance includes a number of actions, such as taking the prescribed drugs, ensuring sufficient sleep, managing stress, and being physically active. Therefore, monitoring self-care means identifying their own clinical changes in terms of new onset or worsening signs/symptoms [7] and addressing these changes with appropriate actions [10].

In this context, health literacy (HL), i.e., “the degree to which individuals have the ability to obtain, process and understand the information and basic health services necessary to make appropriate health decisions” [11], seems to be a crucial determinant of public and individual health, as well as self-care [12,13]. For instance, individuals with low health literacy demonstrated an increased use of healthcare and a greater need for hospitalization than people with high health literacy [14]. Health literacy has also been repeatedly linked to medication adherence, although results on this issue remain inconclusive [15,16,17,18,19,20]. In fact, the literature highlights conflicting results in this regard; if some studies reported significant positive associations between health literacy and medication adherence [21,22,23,24,25], others found significant negative associations [26,27].

In relation to what has been outlined, it could be useful to analyze people’s interest in the issues above. Poor medication adherence leads to increased healthcare costs resulting from poor disease management and the need for hospital admissions, representing a substantial risk factor for morbidity, mortality, and poor quality of life [6,28,29,30]. Elements such as self-care and health literacy could impact these outcomes [8,9]. Assessing how much people are informed about it and how much importance is given to these topics could help organizations define effective health policies.

To analyze and predict human behavior, the study of internet data in the latter years has become an integral part of health informatics, leading to the development of infodemiology [31]. Infodemiology could help us estimate the interest in medication adherence, detect correlations with health literacy and self-care, and evaluate possible changes over time. The presence of changes over time in interest toward health-related issues nowadays becomes relevant, considering, in particular, that the COVID-19 pandemic has led to an unprecedented worldwide increase in internet searches on health topics. Indeed, the Internet has allowed people to stay connected and quickly obtain information related to health. However, not always such information is verifiable and supported by scientific evidence.

For instance, while social media can facilitate the dissemination of accurate health information and connect individuals, it also presents a platform for spreading harmful misinformation, as occurred with anti-vaccine sentiments. In this case, social media platforms have facilitated the propagation of vaccine hesitancy, with exposure to vaccine-critical websites and blogs negatively influencing attitudes toward vaccination. Of note, the credibility of information sources on social media is challenging to ascertain, and to address these concerns, healthcare providers should leverage social media platforms to communicate directly with patients and counter vaccine misinformation. Moreover, structural changes within social media networks, such as flagging harmful misinformation and promoting content from public health agencies, are crucial to effectively combating vaccine misinformation [32,33].

Future research should clarify whether this increased interest has also resulted in a greater ability to filter and analyze the data obtained [34,35,36,37,38].

The primary aim of this study was to describe the global community’s interest in medication adherence and its relationship with health literacy and self-care using Google Trends. The secondary aim was to evaluate the impact of the COVID-19 pandemic on interest in these issues.

## 2. Materials and Methods

### 2.1. Data Collection

We conducted a search through Google Trends, a tool that allows us to know the search frequency on web search engines for a specific term. Google Trends uses revealed and undeclared user preferences [39], allowing one to obtain information that would otherwise be difficult to collect. The tool enables the analysis of sensitive topics as web searches are performed anonymously.

Google Trends allows the user to access, download, and analyze the number of all web queries on a certain topic on the Google Search website. In particular, the tool calculates the proportion of searches for a user-specified term among all searches performed on Google and provides a relative search volume (RSV), i.e., the query share of a particular term for a specified geographical area and period normalized by the highest query share of that term in the time series [40]. The RSVs vary in a range ranging from 1 to 100, where a value of 100 indicates peak popularity and 0 indicates very low search volumes [40,41]. If the RSV is equal to 50, it indicates that half of the searches on a specific term were carried out compared to the highest volume of searches in that period.

In addition, the RSV can compare relative interest between different geographic areas, adjusting for population size and allowing comparison between the most populated and the least populated areas. In order to compare the frequency of searches between different countries, data are standardized by dividing by the total number of searches in that geographical area and in a specific time interval [42]. To compare relative searches’ popularity, each data point is divided by the total searches performed in a specific geographic area and time interval. Otherwise, places with the highest search volume would consistently be ranked highest. By normalizing the search data, Google Trends enables researchers to analyze search interests and facilitate cross-country and cross-language comparisons.

Searches carried out by the same person in a short period of time are automatically excluded from Google trends, thus avoiding overestimating the results. Search queries in Google Trends are defined by a term or topic. The database does not store information about any user’s identity, Internet Protocol address, or specific physical location. In addition, all original web search logs older than 9 months are anonymous in accordance with Google’s privacy policy.

This study analyzed Google Trends in the last decade, from June 2012 to June 2022. In order to assess the influence of the COVID-19 pandemic on interest in the topics under investigation, we conducted a comparative analysis between two specific periods: the pre-pandemic and the pandemic periods. The pre-pandemic period refers to the interval from 1 June 2012 to 10 March 2020, encompassing the period leading up to the official declaration of the COVID-19 pandemic. This period serves as a reference point to establish the baseline level of interest in the studied topics before the emergence of the pandemic.

The identified pandemic period spans from 11 March 2020, which coincides with the WHO’s declaration of the COVID-19 pandemic [43], until 30 June 2022, which marks the end of the designated decade considered in the present study. Of note, the upper limit of the pandemic period evaluated does not refer to the end of the COVID-19 pandemic, which is still ongoing despite not being a global emergency since May 2023 [44].

By examining the data from these distinct periods, we aimed to identify any significant changes or shifts in interest related to the topics under investigation in response to the pandemic.

We aimed to encompass a comprehensive global perspective on the topic of interest by including all countries in our analysis. To ensure a diverse representation of populations and cultural contexts, we conducted data collection and analysis worldwide. However, we must acknowledge that not all countries provided significant data that could be effectively presented and discussed in our manuscript. Consequently, we decided to focus on countries that demonstrated substantial trends related to our research objectives.

We explored Google Trends data for search terms in 3 languages, i.e., English, Italian, and Spanish.

Three main keywords were used for our search: medication adherence (medication adherence, treatment adherence, aderenza terapeutica, adherencia terapéutica), health literacy (health literacy, alfabetizzazione sanitaria, literatura saludable), and self-care (self-care, autocura, autocuidado). We applied to the search a filter to select only the results in the health area.

Some sensitivity analyses were performed to ensure the accuracy and robustness of our search strategy. By systematically varying the search terms, incorporating synonyms and related keywords, and evaluating the impact of these modifications on the results, we assessed the stability of our findings. For example, in the case of medication adherence, we included related terms such as treatment adherence. This modification allowed us to encompass a broader scope of research on this topic, as both terms refer to the same concept. Additionally, the inclusion of related keywords is crucial for capturing the interconnectedness of concepts. Self-care, for instance, is a multidimensional concept that encompasses various aspects of personal health management. By incorporating the related terms “autocura” in Italian and “autocuidado” in Spanish, we aimed to encompass a broader range of internet searches that explore the different dimensions and perspectives of self-care.

The sensitivity analyses we conducted provided valuable insights, as they revealed that the overall patterns and trends remained consistent across different combinations of search terms. This consistency reinforces the reliability and validity of our results. Moreover, to strike a balance between inclusiveness and specificity, we employed a combination of broad and specific search terms. This approach allowed us to minimize the risk of overlooking important data while focusing on the specific aspects relevant to our study.

### 2.2. Statistical Analysis

The data analysis aimed at showing RSV variations during the observation period and comparing the results by country and between the pre- and during the COVID-19 pandemic. Averages and standard deviations were calculated for query search volumes over the study period.

Google Trends provides RSV values on a relative scale, which allows for comparing search volumes across different periods and regions. By employing average RSVs, we aimed to capture the overall search interest during the study period. This approach helps to smooth out daily fluctuations and provides a more robust representation of the search behavior observed.

Moreover, we evaluated the simple correlation between RSVs related to medication adherence, health literacy, and self-care. The Pearson correlation coefficient expressed the extent to which interest in these topics correlated. For this analysis, considering the non-normal distribution of RSVs, we made a logarithmic transformation of RSV values.

Given that the data did not follow a normal distribution, we employed the Mann–Whitney U-test to compare relative search volume (RSV) values related to self-care before and after the COVID-19 pandemic and determine whether there were significant differences. The Mann–Whitney U-test is a non-parametric test suitable for comparing independent samples that do not conform to a normal distribution. By utilizing this test, we could evaluate the significance of the observed differences between the pre- and during the pandemic periods.

Furthermore, the choice of this specific test is dictated by the fact that the two samples compared in our study are not composed of the same statistical units for which the evolution of a specific phenomenon occurs. Instead, they represent independent observations at two different time points.

The statistical analyses were carried out using IBM^®^ SPSS^®^ Statistics version 26, IBM Corp., Armonk, NY, USA. All tests were two-tailed, and a *p* < 0.05 was considered statistically significant.

## 3. Results

General trends showed that interest in self-care showed the greatest increase over time with a mean RSV of 44 (SD ± 18), followed by health literacy with a mean of 29 (SD ± 6) and medication adherence (mean 29, SD ± 5) (Figure 1). Figure 2 illustrates the temporal pattern analysis of the logarithm of RSVs during the decade analyzed. Monthly RSVs for self-care were persistently higher than those related to medication adherence and health literacy during the observation period and had a relevant increase from 2020 (Appendix B, Table A1). In the bivariate correlation analysis (Figure 3), we found a moderate positive correlation between the RSVs related to medication adherence and health literacy (r = 0.674, *p* < 0.0001), medication adherence and self-care (r = 0.466, *p* < 0.0001), and health literacy and self-care (r = 0.545, *p* < 0.0001).

Concerning the people’s interest in medication adherence, some African countries (e.g., Ethiopia, Ghana, Kenya, and Nigeria) showed the highest RSVs. In New Zealand, Australia, and the United States, people seemed to have the greatest interest in health literacy (RSVs of 100, 95, and 50, respectively), with some African countries (e.g., Kenya, Ghana) and Singapore (47) also reporting a relevant number of RSVs. Conversely, countries like Japan, Brazil, Vietnam, and Mexico demonstrate the lowest interest in this issue. As regards self-care, the highest RSVs were found for Latin American countries, including Chile (100), Colombia (91), Peru (81), Ecuador (79), Guatemala (76), and Nicaragua (47). Instead, Asian and African countries like Sri Lanka, Thailand, and the United Arab Emirates showed the lowest interest in self-care (Figure 4).

We conducted a Mann–Whitney U-test to compare the values of RSVs relative to self-care before (median = 36.00, IQR = 28.00–41.00) and after (median = 69.50, IQR = 62.00–81.50) the outbreak of the COVID-19 pandemic, finding a significant increase following the pandemic (U = 17.00, z = −7.902, *p* < 0.0001). RSVs related to health literacy showed two peaks during the pandemic period, in September 2020 and March 2022, with no statistically significant differences compared with the pre-pandemic period. Lastly, RSVs related to medication adherence showed a continuous trend between the pre- and during the COVID-19 pandemic periods.

Most countries show homogeneous search results for medication adherence, health literacy, and self-care, unlike Latin countries, which presented a stronger interest in self-care (Appendix A).

## 4. Discussion

This study suggests that over the last decades, and especially during the COVID-19 pandemic, people’s interest in self-care has substantially increased more than medication adherence and health literacy. Moreover, our analysis of Google’s trends on these topics underlines that interest in medication adherence is directly associated with health literacy and, to a lesser extent, with self-care.

When evaluating possible geographical differences, we found that Google’s trends for the studied topics varied widely across different countries. For instance, some Latin American countries, such as Chile, Colombia, and Peru, had the highest interest in self-care. Instead, lower interest levels were found in Asian and African countries, such as Sri Lanka, Thailand, and the United Arab Emirates. One possible explanation for the findings observed in Latin America could be the cultural emphasis that has been placed on personal well-being and stress management. In these countries, self-care practices could be seen as a way to promote physical and emotional health, enhance personal relationships, and improve the overall quality of life [45,46]. Additionally, many Latin American countries have experienced significant economic growth in recent years, which may have contributed to a greater awareness of the importance of self-care and wellness. On the other hand, the lowest levels of interest in self-care observed in some Asian and African countries could be due to cultural and socioeconomic factors. In some contexts, self-care may be viewed as the behavior of wealthy people or those with more free time. In addition, access to information about self-care practices and resources may be limited in some areas, mainly rural or low-income communities. Overall, these data suggest that interest in self-care varies widely across different countries and cultures. In this regard, it is important to note that the data from Google Trends only reflects online searches and does not necessarily reflect a given population’s self-care practices or behaviors. Nonetheless, these data provide valuable insights into the public’s interest in and awareness of the importance of self-care worldwide.

As far as interest in medication adherence is concerned, Ethiopia tops the list, and this may be due to the high burden of infectious diseases such as HIV/AIDS, tuberculosis, and malaria in the country [47]. These conditions require strict medication adherence to ensure successful treatment outcomes; therefore, people may be more interested in learning about medication adherence. Some other African countries with an increasing prevalence of non-communicable diseases such as diabetes and hypertension [48], in addition to infectious diseases, showed higher RSVs for medication adherence [49,50]. Similarly, South Africa and Malaysia showed a relatively high value of RSVs on this topic, which may be due to the high prevalence of HIV/AIDS and other infectious diseases in South Africa [51,52,53] and the high prevalence of diabetes and hypertension in Malaysia [54,55,56,57]. Of note, the interest in medication adherence from Google trends seemed not to be very marked in high-income countries, such as the United States, Australia, and the United Kingdom. One hypothesis underpinning this finding is that in these areas, the general population is already aware of the importance of medication adherence and has easier access to healthcare resources, which decreases the need for searching the web for information on the topic. This rationale could not be applied to the results found for Saudi Arabia; in this case, people’s cultural beliefs and practices are more likely to influence their willingness to be informed about medication adherence, but further investigations are needed to verify this issue [58].

In line with the picture described above, investigating how RSVs on health literacy vary across different countries reveals some interesting insights. The geographical areas that ranked at the top of the list were New Zealand and Australia, followed by some African countries and Singapore, which appeared to have a high level of interest in health literacy among their populations. These countries are known to have a high burden of communicable and non-communicable diseases, which requires the population to have a good understanding of health literacy to prevent and manage these diseases [48,59,60,61]. The high interest in health literacy in these countries may also characterize healthcare professionals and policymakers since increasing the empowerment and knowledge of the population to make informed decisions about their health has emerged as a current priority for improving the prevention and management of acute and chronic diseases. Moderate levels of interest in health literacy emerged in the United States, Canada, and the United Kingdom. These countries have well-established healthcare systems and generally high educational and literacy rates. Still, the moderate RSV levels suggest there may be room for improvement in health literacy awareness among people. On the other hand, countries such as Japan, Brazil, Vietnam, and Mexico have very low search volume indices, indicating a lack of interest and awareness in health literacy. This may be due to a scarce emphasis placed on health education and literacy in these countries or limited access to health resources, resulting in a lack of awareness about the importance of health literacy. Interestingly, some countries, such as Saudi Arabia and India, have relatively low search volume indices despite having high disease burdens. This issue may be attributed to cultural beliefs and practices potentially affecting health literacy or limited access to health resources and information [58].

The rise of people improving their medical knowledge using web-based sources provides an opportunity to assess trends and interests in a given issue. Internet data is a powerful tool for understanding human behavior and is increasingly integrated into health informatics research [42]. In this context, infodemiology has become a valuable tool to inform public health workers [31,40,62,63,64,65,66,67,68]. For instance, analysis of web-based behavior and behavioral change could help us better understand and interpret the interaction between patients and healthcare professionals [69].

To our knowledge, this is the first study that analyzes Internet data on medication adherence. Google search query volumes for self-care were higher and have markedly increased since 2020. This tendency could be explained by the COVID-19 outbreak and lockdown, which, especially during the first waves, determined substantial changes in managing chronic diseases due to social distancing and limited access to care. In that context, people’s and healthcare professionals’ interest in self-care could have been raised. In contrast, the RSVs related to medication adherence and health literacy have a constant trend over time and are strongly correlated. Several studies in the current literature have studied the relationship between medication adherence and health literacy. Most reported positive associations between medication adherence and health literacy [70,71,72,73], but some found no significant results [74,75,76]. Specifically, the relationship between medication adherence and health literacy was observed among people with chronic diseases [72], including atrial fibrillation [73], hypertension [70], and individuals participating in a health promotion initiative [77]. Conversely, health literacy was not significantly associated with adherence to medications (but with adherence to lifestyle recommendations, e.g., physical activity, alcohol intake, and a healthy lifestyle) in patients with coronary heart disease [74] and those taking antithrombotic drugs [75]. Moreover, another study on women with osteoporosis showed that self-reported income was the only significant predictor of adherence, not health literacy [76].

Even though the correct intake of drugs is considered a self-care activity, there is still insufficient evidence concerning the link between medication adherence and self-care. Indeed, medication adherence represents a particular aspect of the behavioral patterns involved in self-care, including weight control, a healthy diet, and quitting smoking [78,79]. Unlike other aspects, the dynamics surrounding medication intake involve taking the drug at defined times according to a medical prescription and monitoring its positive and negative effects [80]. A study conducted in Ghana on medication adherence and self-care behaviors among patients with type 2 diabetes mellitus, for example, found that non-adherence to medication was directly associated with low levels of education, younger age, and bad practice of self-care activities [81]. Moreover, knowledge of diabetes also significantly influenced self-care levels. Although this and other works suggest that diabetes may be considered a self-manageable disease [82,83], some authors also underlined that medication adherence could be differentially related to self-care depending on the treatment type. For instance, a study conducted in Malaysia on diabetic patients found moderate levels of medication adherence and high self-care behaviors in following dietary guidelines but not in blood glucose testing [83]. The relationship between these two elements was also analyzed in stroke patients, finding that self-care interventions are more likely to have a greater short-term but not long-term positive effect on medication adherence [84]. Even concerning heart failure, the literature shows that medication adherence and self-care behaviors lead to positive outcomes [85] and reduce hospital readmissions [86,87], although adherence to self-care activities, especially those determining lifestyle changes, is still low [88,89].

Effective self-care for heart failure means that the patient has good health literacy skills and the ability to understand and implement the indications of the healthcare professional [90]. Previous studies have shown the relationship between poor health literacy and greater chances of worse health, poorer self-care skills (including medication adherence), increased hospitalizations, and healthcare expenses [90,91,92]. A systematic review found that low health literacy was significantly associated with poor knowledge of the disease in diabetes, respiratory disease, and other chronic diseases. Moreover, a direct relationship between health literacy and self-efficacy was reported in people with diabetes, cardiovascular disease, and human immunodeficiency virus infection. Finally, having lower beliefs in cardiovascular, musculoskeletal, and respiratory diseases was related to the degree of health literacy [92]. These findings suggest that healthcare professionals should consider implementing strategies to involve people with low health literacy in interventions promoting self-management for chronic diseases [93]. Understanding the relationship between health literacy, health awareness, and the implementation of self-care behaviors is fundamental to identifying interventions that can impact health outcomes [94].

We are aware that our study presents some limitations that should be taken into account when interpreting our results. First, to effectively estimate the tracking of search behavior, Google Trends must be carried out on large populations, and it generally tends to collect data only from high-income countries. However, we found interesting results also for countries with low- or middle-income; therefore, we partly overcame this limitation and were able to explore the extent of the problem in heterogeneous geographical areas. Second, although Google is one of the most widely used search engines on the Internet, we did not take into account other tools that can be used to inform people about a specific topic. Third, we did not question Google Trends in all the different languages but only in English, Spanish, and Italian, and this may have limited the evaluation of the interest in the studied topics, especially in Asian countries. In this regard, we acknowledge the potential impact of language barriers on the results, especially in native languages. To address the example of the native language, search terms in French, German, Portuguese, Korean, and Chinese were included in the data collection process. However, the obtained results from these languages did not meet the necessary threshold for inclusion in the analysis due to low search volumes. These low volumes may be attributed to regional variations in internet usage patterns, search engine preferences, or the popularity of alternative platforms for information retrieval in these languages. Furthermore, it is essential to acknowledge that the range of search terms utilized in these native languages is limited, and it should be noted that there are numerous other languages that were not encompassed within the scope of this study. For overall reasons, we decided not to include these native languages in the analysis to ensure the reliability and quality of the data. Although the omission of these languages may limit the evaluation of interest in the studied topics, we believe that it does not significantly impact the overall findings or conclusions of the study. While recognizing the importance of representing a wide range of languages in research, the focus was on prioritizing data quality and reliability.

In addition to this language-related limitation, it is worth noting that the selection of keywords plays a crucial role in ensuring the accuracy and reliability of research results. In the data collection phase, to identify terms that include the concept of self-care, we focused on using commonly recognized and used terms with global relevance and applicability to ensure the generalizability of our findings. We recognize that there may be specific terms related to self-care in the countries included in our study that are not explicitly considered. However, by focusing on widely recognized terms, we aimed to capture the broader trends and patterns of self-care practices that transcend language and cultural barriers. To accurately define the term “self-care” in Spanish, we consulted a native Spanish speaker, who provided us valuable insights into the specific terminology and cultural connotations associated with self-care in Spanish-speaking countries, thereby enhancing the cultural relevance and accuracy of the study.

One notable limitation is the lack of normalization of the Google Trends results using demographic factors such as average income and age in each country. By incorporating these demographic factors, we may better understand the behaviors related to the topics we investigated. Normalizing the Google Trends data with average income is particularly important to mitigate potential bias resulting from the overrepresentation of high-income countries in the dataset. This normalization approach would allow for a fairer comparison across countries with varying income levels, enabling the discovery of hidden trends and insights that might otherwise be overshadowed. Additionally, normalizing based on age can help identify variations in search behavior patterns among different age groups. Failure to incorporate demographic normalization may lead to overrepresenting certain countries or age groups, potentially limiting the generalizability of our findings.

These aspects should, therefore, be further explored to gain a more nuanced understanding of the relationships between search behavior, demographic factors, and health-related topics.

Of note, in performing infodemiologic analysis, it is often necessary to exclude noisy data, which refers to queries not directly related to the topic under examination. However, this task is not always straightforward and may pose challenges. One approach to address this issue is to narrow the search to a specific category, although in this way we may exclude relevant results that are essential for analysis. In this regard, the lack of a standardized reporting method in this emerging research field leads to ambiguity. For instance, Google Trends can provide data on relative search volumes, search volumes, online search traffic data, and online queries, among other terms. Therefore, further efforts should be made to establish specific guidelines for using Google Trends in social and scientific studies and a unified approach to reporting these data among researchers in the field. Finally, because Google Trends does not record information about the identity of the individual who performs the search or his/her specific location, it was not possible to investigate whether the interest in medication adherence, self-care, or health literacy differed by sociodemographic characteristics (e.g., age, sex, gender, educational, or socioeconomic level) or other relevant medical aspects (e.g., multimorbidity). Targeted observational studies should be implemented to derive information in this regard.

## 5. Conclusions

The study sheds light on the escalating interest among individuals concerning health-related matters. People are increasingly inclined to explore subjects such as medication adherence, which encompasses the degree to which patients comply with their prescribed medication regimens. Health literacy, on the other hand, entails the comprehension and practical application of health-related information, while self-care involves the proactive steps individuals take to maintain and improve their overall well-being. Importantly, the study identifies the COVID-19 pandemic as a catalyst that has significantly propelled interest in self-care practices as individuals have become more invested in prioritizing their health amidst the global health crisis.

By examining the relationship between these three topics, we found a particularly noteworthy correlation between medication adherence and health literacy interests. Additionally, the study advocates for using internet data within the medical field to derive valuable insights into the health profiles of different populations. Through analyzing online search patterns, researchers and healthcare professionals can gain a comprehensive understanding of people’s health concerns, interests, and behaviors. In turn, healthcare systems can disseminate tailored and pertinent health information to a wider audience by collaborating with popular social media channels such as Facebook, Instagram, or Twitter. Overall, infodemiology can therefore be considered an interesting field of research that might contribute to developing predictive models that anticipate health trends and identify potential areas for interventions and preventive measures.

## Figures and Tables

**Figure 1 jpm-13-01090-f001:**
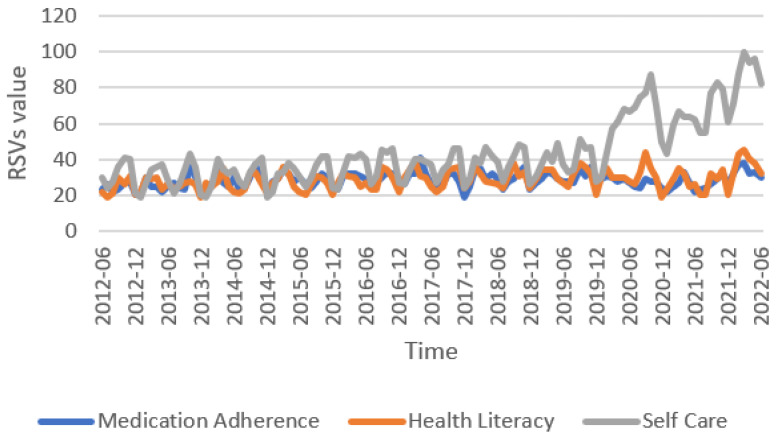
Relative search volume queries related to medication adherence, health literacy, and self-care worldwide in the last decade. Abbreviations: RSVs, Relative search volumes.

**Figure 2 jpm-13-01090-f002:**
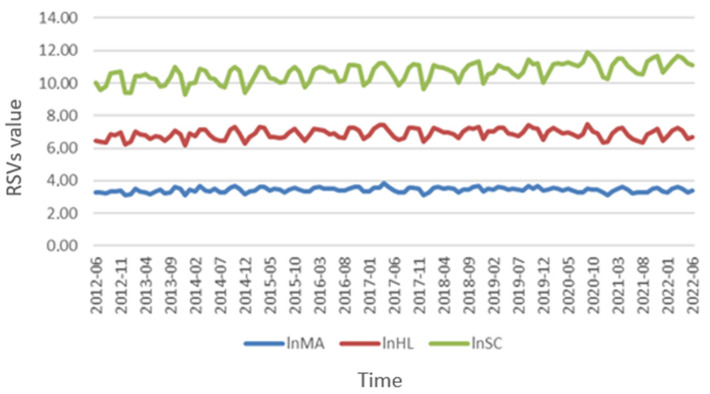
Temporal variation of the logarithm of relative search volumes. Abbreviations: lnHL, logarithm of health literacy searches; lnSC, logarithm of self-care searches; lnMA, logarithm of medication adherence searches; RSVs, Relative search volumes.

**Figure 3 jpm-13-01090-f003:**
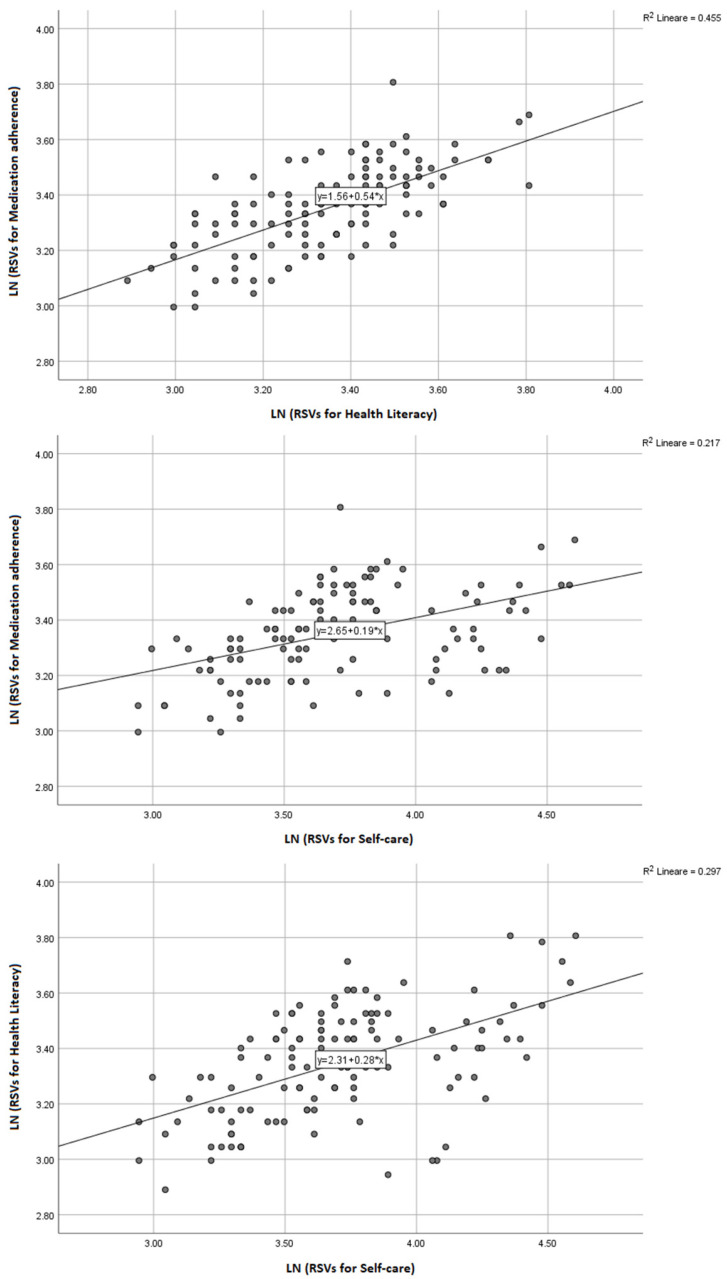
Correlation between the logarithm of relative search volumes for medication adherence and health literacy, medication adherence and self-care, and health literacy and self-care. Notes. The Figures represent the correlations between medication adherence and health literacy, medication adherence and self-care, and health literacy and self-care searches after logarithmic transformation.

**Figure 4 jpm-13-01090-f004:**
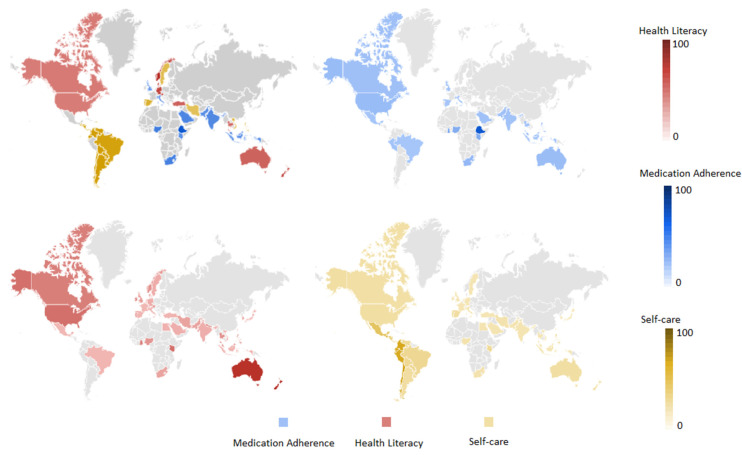
Worldwide distribution of RSVs related to search terms. Abbreviations: HL, health literacy; MA, medication adherence; SC, self-care.

## Data Availability

Data can be made publicly available upon reasonable request.

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
