# Peer review of "Relationship between People’s Interest in Medication Adherence, Health Literacy, and Self-Care: An Infodemiological Analysis in the Pre- and Post-COVID-19 Era"

_jpm, 2023, doi:10.3390/jpm13071090_

Round 1

Reviewer 1 Report

The COVID-19 pandemic has brought unprecedented challenges to the healthcare system, with medication adherence, health literacy, and self-care emerging as crucial areas of focus. In this regard, the study explored the dynamic relationship between these factors and how they evolved during the pandemic. The authors used Google Trends to measure the relative search volumes for medication adherence, health literacy, and self-care from 2012 to 2022. The findings revealed that interest in self-care demonstrated the most significant increase over time, followed by health literacy and medication adherence.

The paper is well-organized and provides a proper introduction to the theme. However, before considering the paper for publication, it is important to seek clarification from the authors on the following points:

- What is the sensitivity of the study results to the search terms used to collect data from Google? The paper presents data from several countries, including Germany, Brazil, and Indonesia, where English, Italian, and Spanish are not native languages.

- Additionally, some of the presented countries may have specific terms for self-care that were not considered in the study.

- The paper is highly interesting; however, the figures are difficult to read.

The English in the text is correct. However, I suggest some modifications to improve the clarity of the message. Here's my suggestion:

While the English in the text is free of major errors, it would be beneficial for authors to avoid writing excessively short paragraphs. This would help to enhance the flow and coherence of the text, making it easier for readers to follow and understand the author's intended message.

Reviewer 2 Report

Review

I would like to express my sincere gratitude to the editor for granting me the opportunity to review such intriguing topics.

Comment 1

At the bivariate correlation analysis (Figure 3), we found a direct relationship between the 157 RSVs related to medication adherence and health literacy (r = 0.674, p <0.0001), medication 158 adherence and self-care (r = 0.466, p <0.0001), health literacy and self-care (r = 0.545, p 159 <0.0001).

I kindly suggest that the author consider determining the level of correlation (r) in order to minimize potential biases, despite the fact that the authors have already presented the correlation (r) and P-value. Nevertheless, it appears that the interpretation of the correlation tends to indicate a low to moderate positive correlation.

Comment 2

Figure 4 displays the world distribution of RSVs related to every single search term. 161 Concerning the peoples interest in medication adherence, some African countries (e.g. 162 Ethiopia, Ghana, Kenya, and Nigeria) showed the highest RSVs. In New Zealand, Aus-163 tralia and United States, people seemed to have the greatest interest in health literacy (RSV 164 of 100, 95 and 50 respectively), with some African countries (e.g. Kenya, Ghana), and Sin-165 gapore (47) also reporting a relevant number of RSVs. Conversely, countries like Japan, 166 Brazil, Vietnam, and Mexico demonstrate the lowest interest in this issue. As regards self-167 care, the highest RSVs were found for Latin American countries, including Chile (100), 168 Colombia (91), Peru (81), Ecuador (79), Guatemala (76), and Nicaragua (47). Instead, Asian 169 and African countries like Sri Lanka, Thailand, and the United Arab Emirates showed the 170 lowest interest in self-care.

I respectfully recommend that the authors explicitly state that the estimated RSVs (Relative Search Volumes) are contingent upon either the total number of search terms or the proportion of search terms relative to the probability of the total population search. Failing to account for the proportionality to the total population could result in countries with larger populations having higher RSVs, potentially skewing the results.

Furthermore, I kindly encourage the authors to provide color labels in Figure 4 to enhance the clarity and comprehensibility of the map.

Comment 3

We conducted a Mann-Whitney U-test to compare the values of RSVs relative to self-172 care before (Median = 36.00, IQR = 28.00 41.00) and after (Median = 69.50, IQR = 62.00 173 81.50) the outbreak of the COVID-19 pandemic, finding a significant increase following 174 the pandemic (U = 17.00, z = -7.902, p <0.0001).

I kindly request that the authors provide clear and precise definitions for the periods before and after the pandemic, as I find the current delineation to be unclear. Defining each period explicitly will help ensure a comprehensive understanding of the study's timeline

Comment 4

Supplementary Figure 1 should be in part of the supplementary, not the manuscript. The author should change to Figure 4 if it needs to remain in the manuscript.

Comment 5

Could the authors give the accurate total numbers of RSVs searching in the study periods

Comment 6

According to “We explored Google Trends data for search terms in 3 languages. i.e. English, Italian and 128 Spanish.

The author discussed  “When evaluating the presence of possible geographical differences, we found that 218 Googles trends for the studied topics varied widely across different countries. For in-219 stance, some Latin American countries, such as Chile, Colombia, and Peru, had the highest 220 interest in self-care. Instead, lower interest levels were found in Asian and African coun-221 tries, such as Sri Lanka, Thailand, and the United Arab Emirates. One possible explanation 222 for the findings observed in Latin America could be the cultural emphasis that has been 223 placed on personal well-being and stress management. In these countries, self-care prac-224 tices could be seen as a way to promote physical and emotional health, as well as to en-225 hance personal relationships and overall quality of life.”

It is important to note that the majority of individuals in Latin America use Spanish or Portuguese, while English is more commonly used in Europe and America. This language limitation should be acknowledged as a major constraint of the study in the initial discussion. Additionally, it should be considered that people in Asia and Africa predominantly use their respective native languages, which may result in search terms being in languages not included in the study.

To address these limitations, I propose that the authors compare RSVs within the same language or focus solely on Latin America, Europe, and North America. Alternatively, they could compare RSVs exclusively in countries where English, Italian, and Spanish are the primary or secondary languages. By doing so, the discussion and conclusion can be adjusted accordingly, accounting for the exclusion of search terms in limited languages.

Overall, it is crucial for the discussion and conclusion to reflect the implications of the study's findings while considering the limitations imposed by the restricted language selection.

Reviewer 3 Report

The paper is interesting, and it sounds that can contribute to the field with valuable information. However, the manuscript needs some modifications. I believe that it is possible the manuscript to be improved and to qualify for publication.

Infodemiology will reveal much of hidden information when we search for answers in typical questionnaires but may be proved as a misleading advisor because you need to evaluate multiple parameters and various normalizations in the efforts to export true conclusion. Big data has many limitations, and we must proceed with great caution.

It would be good idea to further explain and further support with evidence the decision you made to rely solely on google search engine.

You have commented in the study limitations the issues relevant in language choice, the preferred searched terminology but you have to comment or to normalize your results with the mean country or family reported income and age.

Different diseases which have more or less treatment choices and the varying degree of the proportional disease burden in every single country must be taken into account when you interpret your study results. It would be better to try to normalize your results to these parameters either rephrase your study limitations.

The use of English language can be improved in abstract body, and in the  Introduction and material-method section.

Reviewer 4 Report

I read with interest the manuscript presented by Grandieri which describes Relationship between people’s interest in medication adherence, health literacy and self-care pre and post Covid-19 pandemic. The article presents a unique take on how these elements interact and possibly affect each other.

Few comments need addressing:

- Line 42: replace next future with near future.

- The authors provide a comprehensive introduction defining key terms. 

- since COVID-19 was chosen as a point of comparison (pre and post), the mass misinformation associated with COVID-19 should be described, and how such misinformation affected public behaviour such as vaccine hesitancy and resistance (10.3390/tropicalmed7110375) (10.1080/21645515.2020.1780846).

Methods:

- Were all countries included in the analysis? If not, how and what was the rationale for choosing the countries involved in the analysis?

- Statistical analysis is key to yield meaningful results, and should be clearly described in the Methods section. For example: * Why average RSV & SD were used?

* Was the pre and post analysis paired? Please give rationale for your choice.

* Why Mann Whitney test was used? same goes to Pearson.

Results

- Figures should come directly after being referred to in the text. 

- The RSV are written in a difficult-to-read format, but should be written according to standard academic norms. For example, 5.4 (+/- 1.1) [for averages (+/- SD)] or 11 (5.5-13) [for medians (IQR)].

Line 161: Paragraphs should start with a description of a figure. Start by describing your findings, and then refer to your figure.

- Figure 3: Axes should be clearly labeled to faciliate its comprehension.

- Figure 4 is confusing and hard to read. I would suggest to start by clear labels of each part what it represents, and to include a colour gradient to demonstrate the comparison between countries (with clear explanation of the gradient).

- Supplementary figures should be put after the manuscript, not included with the remaining figures.

Discussion:

- The authors presented a decent discussion of their results in light of the current literature. And they have included some limitations to their study.

Limitations of the study:

- In the second limitation, it is stated that Google is the main tool of search, which seems to be an assumption. Either enforce the statement with a reference, or rephrase it as "one of the most widely used search engines".

- In the third limitation, the authors rightly point out the languages used in their study, suggesting asian languages. However, China is included in the current study which would invalidate its inclusion in the current study. To overcome this, I would rephrase the limitation to include other languages, e.g., Arabic which is widely used in the Middle East. 

Moderate editing of English language required

Round 2

Reviewer 4 Report

The authors have done a great job at addressing all my comments, and happy for it to be published as is. Well done!

Author Response

Thank you sincerely for your invaluable comments and feedback on our manuscript. We greatly appreciate the time and effort you invested in reviewing our work. Your insightful suggestions and guidance have been instrumental in shaping the final draft of the manuscript.

We are thrilled to hear that you are satisfied with the revisions we made to address your comments. Your positive assessment and approval for publication mean a great deal to us. We are truly grateful for your kind words of encouragement.

Once again, we extend our heartfelt gratitude for your expertise and dedication in reviewing our manuscript. Your contributions have undoubtedly strengthened the quality of our work. We are honored to have had the opportunity to benefit from your expertise.

Thank you once again for your support and encouragement.